# Global patterns of aegyptism without arbovirus

**Mark F. Olson[1], Jose G. Juarez[1], Moritz U. G. Kraemer[2], Jane P. Messina[3], Gabriel L. Hamer[1]***

**1** Department of Entomology, Texas A&M University, College Station, Texas, United States of America, **2** Department of Zoology, University of Oxford, Oxford, United Kingdom, **3** School of Geography and the Environment, and Oxford School of Global and Area Studies, University of Oxford, Oxford, United Kingdom

* ghamer@tamu.edu

**Data Availability Statement:** All relevant data are within the manuscript and its Supporting Information files.

**Funding:** This publication was supported by Cooperative Agreement Number U01CK000512,

## Abstract

The world's most important mosquito vector of viruses, *Aedes aegypti*, is found around the world in tropical, subtropical and even some temperate locations. While climate change may limit populations of *Ae. aegypti* in some regions, increasing temperatures will likely expand its territory thus increasing risk of human exposure to arboviruses in places like Europe, Northern Australia and North America, among many others. Most studies of *Ae. aegypti* biology and virus transmission focus on locations with high endemicity or severe outbreaks of human amplified urban arboviruses, such as dengue, Zika, and chikungunya viruses, but rarely on areas at the margins of endemicity. The objective in this study is to explore previously published global patterns in the environmental suitability for *Ae. aegypti* and dengue virus to reveal deviations in the probability of the vector and human disease occurring. We developed a map showing one end of the gradient being higher suitability of *Ae. aegypti* with low suitability of dengue and the other end of the spectrum being equal and higher environmental suitability for both *Ae. aegypti* and dengue. The regions of the world with *Ae. aegypti* environmental suitability and no endemic dengue transmission exhibits a phenomenon we term 'aegyptism without arbovirus'. We then tested what environmental and socioeconomic variables influence this deviation map revealing a significant association with human population density, suggesting that locations with lower human population density were more likely to have a higher probability of aegyptism without arbovirus. Characterizing regions of the world with established populations of *Ae. aegypti* but little to no autochthonous transmission of human-amplified arboviruses is an important step in understanding and achieving aegyptism without arbovirus.

## Author summary

The preeminent vector of arboviruses, *Aedes aegypti*, is distributed globally and capable of transmitting deadly pathogens to over half the world's population. While most studies focus on areas where *Ae. aegypti* and human-amplified urban arboviruses such as dengue and Zika viruses are locally established, our study explores the margins of endemicity

funded by the Centers for Disease Control and Prevention (G.L.H.; www.cdc.gov). Its contents are solely the responsibility of the authors and do not necessarily represent the official views of the Centers for Disease Control and Prevention or the Department of Health and Human Services. Additional support came from NIH K01AI128005 (G.L.H.; www.nih.gov). The funders had no role in study design, data collection and analysis, decision to publish, or preparation of the manuscript.

**Competing interests:** The authors have declared that no competing interests exist.

where *Aedes aegypti* can be found, but arboviral illness is rare. These areas where we find environmental suitability for the vector but an absence of established arboviral transmission we term 'aegyptism without arbovirus'. This builds on the long-held observation of 'anophelism without malaria' in which some regions having *Plasmodium*-competent *Anopheles* spp. mosquitoes but not the associated human malaria. This study uses previously published maps to reveal locations with higher suitability of *Ae. aegypti* and low suitability of dengue. Additionally, we analyzed the resulting map of deviations between *Ae. aegypti* and dengue and found significant associations with human population density, infant mortality rate, temperature, and precipitation. To the best of our knowledge, this is the first study to characterize places around the world that exhibit 'aegyptism without arbovirus' which is an important first step in our ongoing battle with human-amplified urban arboviruses.

## Introduction

Over half the world's population lives in areas at risk of human-amplified urban arboviruses transmitted by *Aedes aegytpi* mosquitoes [1]. In addition to chikungunya, yellow fever, and Zika viruses, *Ae. aegypti* is the primary vector for dengue virus which infects an estimated 390 million individuals each year [2], with 100 million of those being symptomatic [3]. While great strides have been made in vector surveillance and control through conventional, biological, and genetic approaches, and vaccine development is ongoing [see 4], dengue transmission is expected to persist and in some regions expand while other regions contract [1].

Many studies have identified environmental, meteorological, and demographic factors related to vector populations and arboviral transmission such as human population density, climate, normalized difference vegetation index (NDVI), and gross domestic product (GDP) [5,6]. More recent research has considered the impact of socio-economic status [7] and urbanization including urban heat islands [8] on risk of increased dengue transmission [9,10]. Understandably, studies tend to be conducted in locations of high endemicity for arboviral disease transmission or where recent outbreaks have occurred. Rarely have studies evaluated landscape influences of *Ae. aegypti* populations or arbovirus transmission in locations representing the margins of endemicity. We recently conducted a study in South Texas where large populations of *Ae. aegypti* occur yet local transmission of human-amplified urban arboviruses is rare, and we discovered high rates of non-human feeding by *Ae. aegypti* [11]. These wasted bites on non-amplification hosts likely reduced $R_0$ for ZIKV limiting local transmission to 10 human cases between 2016–2017. In contrast, Tamaulipas, the Mexican state across the border, reported 16,835 cases in the same period. The lower availability of humans to *Ae. aegypti* and associated utilization of non-human hosts is one of several mechanisms for a phenomenon we term 'aegyptism without arbovirus'; defined as the occurrence of established *Ae. aegypti* populations without endemic human-amplified urban arboviruses. This context is similar to the long-held observation of 'anophelism without malaria' [12,13], where researchers in the 1920s started to notice and understand the mechanisms of some regions having *Plasmodium*-competent *Anopheles* spp. mosquitoes but not the associated human malaria. The objective of this study is to explore the global patterns of environmental suitability for *Ae. aegypti* and dengue to characterize the deviations in these predictions. We addressed this objective by developing a map predicting a gradient ranging from higher environmental suitability for *Ae. aegypti* but low suitability for dengue to the other end of the spectrum where areas have similar and higher suitability for both *Ae. aegypti* and dengue. Our analysis is based upon previously

published data estimating global environmental suitability for *Ae. aegypti* [14] and dengue [1]. We used these suitability maps projected to 5 km$^2$ grids to then further calculate deviations in *Aedes aegypti* and dengue suitability. We then identify environmental, meteorological, and demographic factors associated with this gradient in the deviation between *Ae. aegypti* and dengue suitability to explore the social-ecological factors driving aegyptism without arbovirus.

## Materials and methods

### Deviation between the probability of occurrence of *Aedes aegypti* and dengue

This study utilized the 2015 global probability of occurrence for *Ae. aegypti* based on a mosquito database and environmental variables predicting their global distribution [14]. We also used the 2015 global probability of dengue occurrence which was based on an ecological niche model of human cases to predict environmental suitability [1]. It is important to note that these maps are predictions of environmental suitability, not occurrence of *Aedes aegypti* or dengue. To compare the global pattern of *Ae. aegypti* and dengue suitability we performed raster calculations in QGIS (version 3.10.1-A Coruña). Both the *Ae. aegypti* environmental suitability map and the global probability of dengue suitability are at 5 km$^2$ resolution. We removed all cells where either *Ae. aegypti* or dengue suitability were < 0.1 to filter out locations where environmental suitability for *Ae. aegypti* or dengue virus is extremely low (e.g. Greenland and Arctic locations). To create a map that illustrates where *Ae. aegypti* and dengue deviate spatially, we generated an initial raster that calculated "*Ae. aegypti*" minus "dengue". This procedure removed all pixels where an interaction between *Ae. aegypti* and dengue did not occur. This resulting *Ae. aegypti* minus dengue raster ('Uncorrected deviation layer') produced one end of the spectrum with a suitable environment for *Ae. aegypti* but low suitability for dengue and the other end of the spectrum included an equal suitability for both *Ae. aegypti* and dengue. The problem with this later end was that areas of the world with near zero suitability for both *Ae. aegypti* and dengue were indifferent from areas with high suitability for *Ae. aegypti* and dengue. To account for this, we created seven unique raster's that would incrementally remove areas with lower dengue environmental suitability according to the gradient levels in Table 1. These rasters were merged to develop an image that encompasses the deviation between *Ae. aegypti* probability of occurrence and dengue environmental suitability. Briefly, to create Level 1, we performed the raster calculation: ("Uncorrected deviation layer" ≥ -0.5) AND ("Dengue 2015 filtered" ≥ 0.5). This level represents areas with similar and high

**Table 1. Correction to the deviation between *Ae. aegypti* and dengue map by clipping out respective areas with a lower probability of dengue environmental suitability.**

| Level | Uncorrected deviation in *Ae. aegypti* and dengue | Clip areas ≤ these values for the probability of dengue suitability | Corrected deviation in *Ae. aegypti* and dengue (range) | Description |
|---|---|---|---|---|
| 1 | -0.5 | 0.8 | - 0.5–0.19 | Remaining cells only have higher dengue suitability |
| 2 | -0.35 | 0.75 | -0.35–0.24 | |
| 3 | -0.2 | 0.7 | -0.20–0.27 | |
| 4 | -0.05 | 0.65 | -0.05–0.32 | Remaining cells have medium-higher dengue suitability |
| 5 | 0.1 | 0.6 | 0–0.36 | |
| 6 | 0.25 | 0.55 | 0–0.41 | |
| 7 | 0.4 | 0.5 | 0–0.48 | Remaining cells have lower-higher dengue suitability |

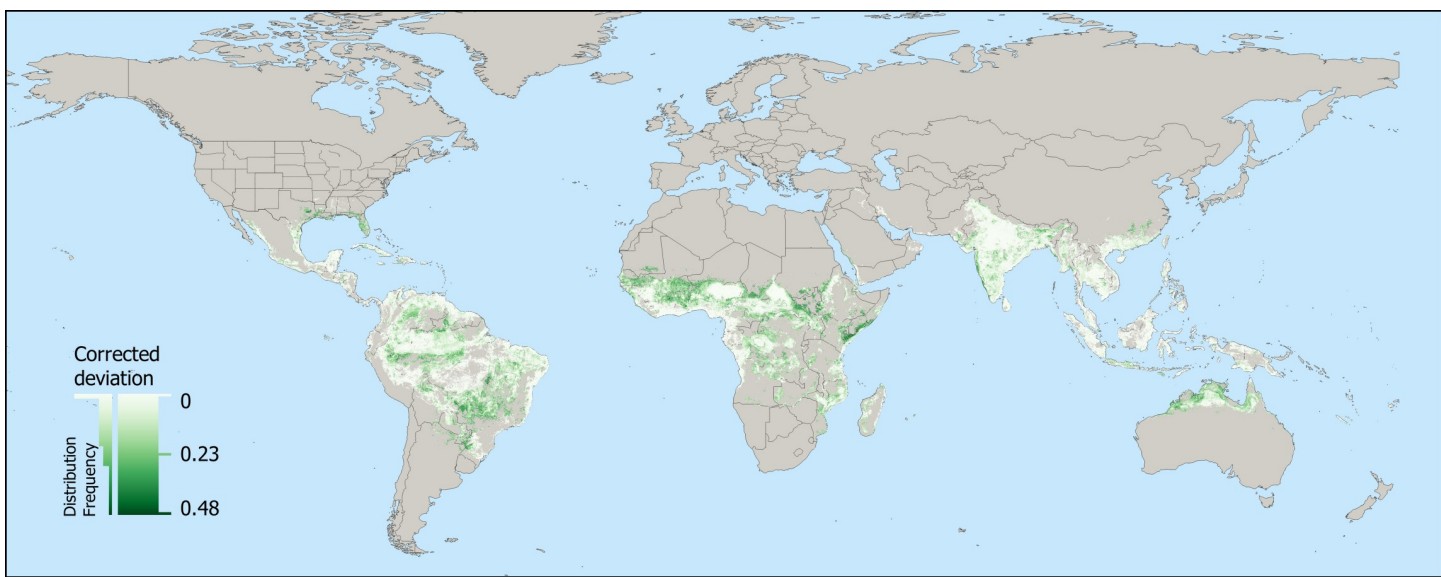

**Fig 1. Deviation between *Ae. aegypti* and dengue environmental suitability.** Green indicates areas where *Ae. aegypti* is likely to be found, but the environment is not considered suitable for dengue transmission (e.g. Southern United States, Northern Argentina, Northern Australia). White indicates areas where the environmental suitability of *Ae. aegypti* and dengue is similar and higher. Inset histogram provides distribution of the corrected deviation values. The map was created by the author using QGIS 3.10 (https://qgis.org/en/site/) with public domain map data from Natural Earth (https://www.naturalearthdata.com/downloads/50m-physical-vectors/) and U.S. Geological Survey (https://woodshole.er.usgs.gov/pubs/of2005-1071/data/background/us_bnds/state_boundsmeta.htm).

environmental suitability of both *Ae. aegypti* and dengue. This same procedure was used to create the remaining levels in Table 1. Because our focus is on aegyptism without arbovirus, we filtered the deviation raster to only include values $\geq 0$ (to exclude areas where dengue environmental suitability was greater than *Ae. aegypti*) (Fig 1).

## Socio-ecological patterns in the deviation between *Ae. aegypti* and arbovirus

To identify environmental, meteorological, and demographic factors relating to the deviations between *Ae. aegypti* probability of occurrence and dengue environmental suitability we gathered several global datasets. Human population density maps and subnational infant mortality rates, both of 2015, were obtained through NASA's SEDAC website [15]. Infant mortality rate is defined as the number of children who die before their first birthday per 1000 live births. Infant mortality rate (IMR) is often used as an indicator for poverty [16] and dengue infection during pregnancy has been linked to increased risk of infant mortality, among other adverse health outcomes [17]. IMR data was available from 234 countries, with 143 of those countries reporting subnational units at the 30 arc-second (approximately 1 km$^2$) resolution [18]. A global map of total gross domestic product (GDP) per capita data at 30 arc-sec resolution for 2015 was obtained from Kummu et al. [19]. Total GDP per 5 x 5 km$^2$ cell was estimated by multiplying per capita GDP by gridded human population data [19]. Global precipitation and temperature rasters at 30 arc-sec spatial resolution were obtained from worldclim.org [20]. These rasters represent average monthly data from 1970 to 2000 and are separated by month. We combined the 12 monthly rasters to create one annual mean temperature raster, and a cumulative annual precipitation raster. While combining the seasonal data into an annual metric loses the opportunity to investigate temporal heterogeneity, we did this to facilitate this exploratory analysis on a global scale. We hypothesize that temperature and precipitation will

have an inverse relationship, where higher annual average temperatures and higher cumulative rainfall will be correlated with a lower deviation value on the scale of equal and greater probability of *Ae. aegypti* occurrence without dengue environmental suitability. We also hypothesize that elevation will be positively correlated with aegyptism without arbovirus. Cells with missing values were removed across all layers before performing the analysis.

We used a gradient boosting machine (GBM) approach with a Gaussian distribution to evaluate how the corrected deviation probability of *Ae. aegypti* and dengue is influenced by human population density, temperature, precipitation, IMR, GDP and elevation. Regression trees were fitted using a learning rate of 0.001, 5-fold cross validation and 10,000 trees to minimize the mean squared error (MSE) loss function [21]. Each tree was iteratively improved using a stepwise manner reducing the variation in the response variable. Models were generated using the "gbm" package in R [22]. Subsequently, we used generalized additive models (GAM) for count data (Poisson) to determine which effect variables (human population density, temperature, precipitation, IMR, GDP and elevation) best explain the variation of the corrected deviation probability. Smoothing terms were evaluated based on their estimated degrees of freedom (edf). We used the adjusted $R^2$ value to determine the best fit model structure. All statistical analyses were conducted in R version 3.5.1 [23] using RStudio version 1.1.456 [24]. To import and analyze rasters in R, we utilized the packages of 'raster', 'dplyr', 'mgcv', and 'ggplot2' [25].

## Results

### Deviation between the probability of occurrence of *Aedes aegypti* and dengue environmental suitability

A map was generated showing global deviations between the probability of *Ae. aegypti* and dengue habitat suitability (Fig 1). Values range from 0 (equal and higher environmental suitability for both *Ae. aegypti* and dengue) to 0.48 (higher suitability of *Ae. aegypti* with low suitability of dengue) with a mean of 0.07 (residual standard error: 0.09). For example, a 5 x 5 km$^2$ area that has a 0.78 probability of occurrence for *Ae. aegypti* but only a 0.3 environmental suitability for dengue would have a deviation value of 0.48. The mean deviation value for South Africa, United States, and Australia is 0.18, 0.16, and 0.13, respectively. Locations with a lower probability of aegyptism without arbovirus include Mexico, Thailand and Guatemala which have a mean deviation value of 0.04, 0.02 and 0.01, respectively. We report the mean deviation values for each country in S2 Table which range from 0 to 0.27.

### Socio-ecological patterns in the deviation between *Ae. aegypti* and dengue

The gradient boosting machine (GBM) full model resulted in an MSE of 0.084. The independent variable contributing the most to explaining the variation in the dependent variable was human population density (38.475) and the variable with least relative influence was elevation (0.067). After removing the elevation data, GBM was conducted again on the remaining variables in a stepwise fashion (Table 2). A generalized additive model revealed model 3 to have the best-fit with an $R^2$ value of 0.152 (Table 3). Statistically significant effects were found between the corrected deviance raster and human population density, IMR, temperature, and precipitation as the smooth terms. The human population density layer had a range of 0 to 119,921 persons per km$^2$ and a mean of 135.93 (residual standard error: 0.09) persons per km$^2$. (Table 4).

The subnational IMR ranged from a low of 0.24 to a high of 142.93 and a mean of 35.63 (infant deaths per 1,000 live births) (residual standard error: 0.09). Using human population

**Table 2. Gradient Boosting Machine (GBM) to determine best-fit model.** Abbreviated variable names include human population density (pop), gross domestic product (gdp), infant mortality rate (imr), annual mean temperature (temp), annual cumulative precipitation (prec), elevation (elev).

| Dependent variable | Independent variables | Greatest relative influence (value) | Least relative influence (value) |
|---|---|---|---|
| amd | pop, gdp, imr, temp, prec, elev | pop (38.475) | elev (0.067) |
| amd | pop, gdp, imr, temp, prec | pop (38.460) | gdp (2.276) |
| amd | pop, imr, temp, prec | temp (44.382) | imr (10.905) |
| amd | pop, temp, prec | temp (48.580) | prec (22.376) |
| amd | pop, temp | temp (68.744) | pop (31.256) |

density, temperature and precipitation as smoothing terms, the parametric coefficient for IMR was 3.002e-04 (± 3.067e-06 SE; p < 0.001). Mean annual temperatures ranged from 6.98˚C to 31.21˚C throughout the range covered by the deviation raster, with a mean of 25.31˚C (residual standard error: 0.09 on 1,207,056 degrees of freedom). The parametric coefficient for temperature was 1.347e-03 (± 4.233e-05 SE; pr (>|t|) = <2e-16). Precipitation had a range of 4 to 9,083 mm rainfall and a global mean of 1,550.18 mm (residual standard error: 0.09 on 1,207,195 degrees of freedom). The parametric coefficient for precipitation, using human population density, temperature and IMR as smoothing terms, was -1.530e-05 (± 1.127e-07 SE; pr (>|t|) = <2e-16).

## Discussion

*Aedes aegypti* has proliferated in urban areas around the globe in the last century. While ubiquitous in many tropical and subtropical urban areas, some locations infested with *Ae. aegypti* do not exhibit high levels of human-amplified urban arboviral transmission as in other areas. This study built on previous studies mapping the global suitability of *Ae. aegytpi* and dengue to generate a map of deviation values including the observation of aegyptism without arbovirus. We produced a global map showing this gradient from high suitability for *Ae. aegypti* but low suitability for dengue to the other end of the spectrum where areas have similar and higher suitability for both *Ae. aegypti* and dengue. We show that some countries on the margins of endemicity of human-amplified arboviruses have a higher deviation value compared to highly endemic countries. For example, the U.S. and Argentina, both countries with occasional autochthonous transmission of dengue virus [26–28] have mean deviation values of 0.16 and 0.18, respectively (S2 and S3 Figs). These higher values along this spectrum are more representative of aegyptism without arbovirus. This is also corroborated by empirical data showing that even in areas with high abundances of *Ae. aegypti*, low human feeding diminishes the risk of Zika virus transmission [11,29]. Likewise, two major urban centers of Kenya exhibit higher values of aegyptism without arbovirus while Mombasa, a coastal city in the same country has frequent dengue epidemics (S4 Fig) [29]. Countries highly endemic for dengue, such as Honduras and Thailand, have mean deviation values of 0.038 and 0.023, respectively, which are values representing the regions with higher suitability for both *Ae. aegypti* and dengue. We

**Table 3. Results of Generalized Additive Model (GAM).** Family: gaussian; link function: identity.

| Model | Formula | Adjusted R² | Deviance explained |
|---|---|---|---|
| 1 | amd ~ s(pop) + s(gdp) + s(imr) + s(temp) + s(prec) + s(elev) | 0.115 | 11.5% |
| 2 | amd ~ s(pop) + s(gdp) + s(imr) + s(temp) + s(prec) | 0.107 | 10.7% |
| 3 | amd ~ s(pop) + s(imr) + s(temp) + s(prec) | 0.152 | 15.2% |
| 4 | amd ~ s(pop) + s(temp) + s(prec) | 0.138 | 13.8% |
| 5 | amd ~ s(pop) + s(temp) | 0.113 | 11.3% |

**Table 4. Results of Generalized Additive Model (GAM) for Model 3.** Family: gaussian; link function: identity. (Formula: amd_r ~ s(pop_r) + s(imr_r) + s(temp_r) + s (prec_r); n = 1,190,702).

| Parametric coefficients: | | | | |
|---|---|---|---|---|
| | Estimate | Standard Error | t-value | pr (>|t|) |
| (intercept) | 7.223e-02 | 8.197e-05 | 881.15 | <2e-16*** |
| | | | | |
| **Approximate significance of smooth terms:** | | | | |
| | edf | F | p-value | |
| s(pop) | 9.000 | 5888 | <2e-16*** | |
| s(imr) | 8.998 | 1981 | <2e-16*** | |
| s(temp) | 8.988 | 10640 | <2e-16*** | |
| s(prec) | 8.997 | 2597 | <2e-16*** | |

*** < 0.001

identified a significant association between human population density and the deviation in environmental suitability of *Ae. aegypti* and dengue. Locations with higher deviation values had lower human population densities. This means that regions of the world with aegyptism without arbovirus are more likely to be lower human population densities compared to regions with more equal and higher probabilities of *Ae. aegypti* and dengue. It was surprising to see that GDP did not have a significant effect on the deviation values. Åström et al. modeled various scenarios of dengue distribution according to climate and socioeconomic change, finding a beneficial, protective effect from increasing GDP [30]. Locations with higher GDP would presumably have better access to piped water, screened windows and possibly air conditioning, factors which could reduce arboviral transmission [31]. In addition to GDP, Kummu et al. also mapped a human development index (HDI) which is composed of the achievement of several key development indicators, and this may be a better predictor of deviation. Interestingly, the deviation values for aegyptism without arbovirus were positively correlated to infant mortality rates. We expected to see higher deviation values representing aegyptism without arbovirus in places with lower IMR, but this wasn't the case. One potential explanation is reporting bias with some low-income areas having higher dengue burdens than what are reported. For example, Africa has a wide variety of common febrile illnesses with varying etiology, thus a case of dengue fever could be inadvertently misdiagnosed as malaria, especially in places where testing is less than rigorous or non-existent [32]. Regions with notoriously high IMR, but where dengue is underreported could therefore appear to have higher presence of *aegypti* without arbovirus.

Recent studies suggest that climate change, while limiting expansion of *Ae. aegypti* in some locations, will likely increase the risk of human exposure in other areas like North America, Australia and Europe [33,34]. Certainly, temperature plays an important role in its propagation [35]. Interestingly, our study found a significant relationship between temperature and aegyptism without arbovirus, where higher average annual temperatures were associated with higher suitability for *Ae. aegypti* and lower suitability of dengue. This pattern is based on average yearly temperatures and seasonality and diurnal temperature fluctuations were not considered. Carrington et al. found greater potential for dengue virus transmission in *Ae. aegypti* exposed to large diurnal fluctuations at lower mean temperatures [36]. Further study on the effects of temperature on aegyptism without arbovirus is needed. Precipitation is also a main driver of *Ae. aegypti* populations as a water source is necessary for oviposition. We observed a significant effect on deviation where lower average precipitation was associated with higher

probability of *Ae. aegypti* without arbovirus disease. It is interesting to note, however, that many locations with less than 100 mm per year in rainfall were still considered highly suitable for *Ae. aegypti*. Perhaps places with little to no rainfall such as Phoenix, Arizona, are still capable of maintaining high populations of *Ae. aegypti* due to prolific use of water in the urban landscape and abundant container habitat [37].

The complex nature of dengue transmission requires competent mosquito vectors and viremic and susceptible humans to initiate and sustain transmission. This current study does not explore additional factors that could influence the abundance of *Ae. aegypti* and the probability of local transmission of dengue virus. For example, the endophilic behavior and propensity to feed on humans of *Ae. aegypti* is known to vary [38,39]. Some dengue endemic settings have high abundance of indoor populations [40] but in other less endemic areas, outdoor *Ae. aegypti* populations are larger than indoor populations [41]. Also, this study does not consider heterogeneity in virus importation by humans or human herd immunity. Some regions with abundant *Ae. aegypti* have frequent importation of viremic humans helping to initiate local transmission [42]. Heterogeneity in human herd immunity to dengue serotypes is also a factor informing probability of dengue transmission [43], a factor that we have not considered. While we are pointing out regions of the world with higher suitability of *Ae. aegypti* and lower suitability of dengue virus we also acknowledge *Ae. aegypti* is not the only vector for dengue virus. Multiple studies have documented dengue virus transmission in the absence of *Ae. aegypti* and instead incriminate the Asian tiger mosquito, *Ae. albopictus* as a secondary vector [44–47]. A future study could take a similar approach to identifying global patterns of dengue disease in the absence of *Ae. aegypti* to help provide more evidence of transmission by other vector species.

Our analysis is built upon predictions of environmental suitability of *Ae. aegypti* [14] and dengue [1], which introduces sources of error and uncertainty. For example, Messina et al. [1] global predictions of dengue includes high risk in regions such as Arkansas, USA, with values around 0.87 (range of 0–1). There is no documented autochthonous transmission of any human-amplified arbovirus in Arkansas in the last two centuries [48,49]. As a result of this model's prediction, our deviation map includes values in Arkansas from 0–0.15, that would falsely indicate that this region has both similar and high levels of *Ae. aegypti* and dengue. These anomalies likely occur elsewhere in the world with these deviation value predictions, especially in developing countries where differential diagnosis of febrile illness in humans is less common. At the global scale, our deviation map identifies regions around the world on the margins of arboviral endemicity but where the environment is suitable for *Ae. aegypti*. However, at a finer resolution (e.g. at the county or city level), one can find deviation values that don't reflect updated data documenting *Ae. aegypti* and human dengue cases. In the future, this same analysis could be done with improved *Ae. aegypti* and human case data for dengue or other arboviral diseases at a finer spatial scale.

In conclusion, our study identified several focal points around the globe which appear to exhibit this phenomenon of aegyptism without arbovirus. Parts of South America, Africa, South Europe, and North Australia appear to exhibit this same phenomenon that we find in the United States. While *Ae. aegypti* is found in all of these locations and even expanding in many areas, vector presence does not unequivocally translate to the transmission of human-amplified urban arboviruses such as dengue. A suite of factors such as *Ae. aegypti* vector competence, utilization of humans as hosts, and human social practices reducing contact with mosquitoes are likely to influence the risk of arbovirus transmission. Further research to elucidate the underlying mechanisms which facilitate aegyptism without arbovirus is warranted. The knowledge gained from this research will help guide scientists, public health officials and policy makers in our ongoing battle against mosquito-borne viruses.

## Supporting information

**S1 Fig. Deviation between *Ae. aegypti* probability of occurrence and dengue environmental suitability, zoomed in on North America, South America, South Europe and North Africa, Africa, and Australia.** The map was created by the author using QGIS 3.10 (https://qgis.org/en/site/) with public domain map data from Natural Earth (https://www.naturalearthdata.com/downloads/50m-physical-vectors/) and U.S. Geological Survey (https://woodshole.er.usgs.gov/pubs/of2005-1071/data/background/us_bnds/state_boundsmeta.htm).
(TIF)

**S2 Fig. Deviation between *Ae. aegypti* probability of occurrence and dengue environmental suitability for the Southern United States.** The map was created by the author using QGIS 3.10 (https://qgis.org/en/site/) with public domain map data from Natural Earth (https://www.naturalearthdata.com/downloads/50m-physical-vectors/) and U.S. Geological Survey (https://woodshole.er.usgs.gov/pubs/of2005-1071/data/background/us_bnds/state_boundsmeta.htm).
(TIF)

**S3 Fig. Deviation between *Ae. aegypti* probability of occurrence and dengue environmental suitability for Northern Argentina, Paraguay, and Southern Brazil.** The map was created by the author using QGIS 3.10 (https://qgis.org/en/site/) with public domain map data from Natural Earth (https://www.naturalearthdata.com/downloads/50m-physical-vectors/) and U.S. Geological Survey (https://woodshole.er.usgs.gov/pubs/of2005-1071/data/background/us_bnds/state_boundsmeta.htm).
(TIF)

**S4 Fig. Deviation between *Ae. aegypti* probability of occurrence and dengue environmental suitability for Kisumu, Nairobi and Mombasa, Kenya.** The map was created by the author using QGIS 3.10 (https://qgis.org/en/site/) with public domain map data from Natural Earth (https://www.naturalearthdata.com/downloads/50m-physical-vectors/) and U.S. Geological Survey (https://woodshole.er.usgs.gov/pubs/of2005-1071/data/background/us_bnds/state_boundsmeta.htm).
(TIF)

**S1 Table. Data sources for the global rasters used in this paper.**
(DOCX)

**S2 Table. Statistical summary of *Ae. aegypti* minus dengue deviation, by country.** White indicates the lower end of the spectrum, where *Ae. aegypti* occurrence and risk of dengue is nearly equal and high, and green represents the other end of the spectrum where *Ae. aegypti* can be found without dengue. Countries with 5 or fewer cells (5 km$^2$) were removed from the table for brevity.
(DOCX)

## Acknowledgments

We thank Sarah Hamer and Micky Eubanks for providing input that helped improve the analysis and interpretation.

## Author Contributions

**Conceptualization:** Gabriel L. Hamer.

**Data curation:** Mark F. Olson, Moritz U. G. Kraemer, Jane P. Messina.

**Formal analysis:** Mark F. Olson, Jose G. Juarez.

**Methodology:** Mark F. Olson, Jose G. Juarez, Jane P. Messina, Gabriel L. Hamer.

**Project administration:** Gabriel L. Hamer.

**Resources:** Moritz U. G. Kraemer, Jane P. Messina.

**Supervision:** Gabriel L. Hamer.

**Visualization:** Jose G. Juarez.

**Writing – original draft:** Mark F. Olson.

**Writing – review & editing:** Jose G. Juarez, Moritz U. G. Kraemer, Jane P. Messina, Gabriel L. Hamer.

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
