## [Decision Letter · Decision Letter 0]

16 Feb 2021

Dear Dr. Hamer,

Thank you very much for submitting your manuscript "Global patterns of aegyptism without arbovirus" for consideration at PLOS Neglected Tropical Diseases. As with all papers reviewed by the journal, your manuscript was reviewed by members of the editorial board and by several independent reviewers. In light of the reviews (below this email), we would like to invite the resubmission of a significantly-revised version that takes into account the reviewers' comments. 

We cannot make any decision about publication until we have seen the revised manuscript and your response to the reviewers' comments. Your revised manuscript is also likely to be sent to reviewers for further evaluation.

Sincerely,

Mariangela Bonizzoni

Associate Editor

Pedro Vasconcelos

Deputy Editor

Reviewer's Responses to Questions

**Key Review Criteria Required for Acceptance?**

**Methods**

-Are the objectives of the study clearly articulated with a clear testable hypothesis stated?

-Is the study design appropriate to address the stated objectives?

-Is the population clearly described and appropriate for the hypothesis being tested?

-Is the sample size sufficient to ensure adequate power to address the hypothesis being tested?

-Were correct statistical analysis used to support conclusions?

-Are there concerns about ethical or regulatory requirements being met?

Reviewer #1: See my overall review in the general comments

Reviewer #2: Methods are appropriate.

**Results**

-Does the analysis presented match the analysis plan?

-Are the results clearly and completely presented?

-Are the figures (Tables, Images) of sufficient quality for clarity?

Reviewer #1: See my overall review in the general comments

Reviewer #2: Results are clear.

**Conclusions**

-Are the conclusions supported by the data presented?

-Are the limitations of analysis clearly described?

-Do the authors discuss how these data can be helpful to advance our understanding of the topic under study?

-Is public health relevance addressed?

Reviewer #1: See my overall review in the general comments

Reviewer #2: Yes.

**Editorial and Data Presentation Modifications?**

Reviewer #1: See my overall review in the general comments

Reviewer #2: Minor Revision.

**Summary and General Comments**

Reviewer #1: This paper describes a modeling analysis of the differential between projected Ae aegypti populations and a model of the likelihood of dengue transmission in an attempt to investgate why some areas with the vector lack dengue. This is an important topic as the primary vector of dengue, Ae aegypti, is spreading to new areas. In particular, parts of the southwestern United States including urban areas in California and Arizona, have recently been infested with this mosquito. And it appears as though the populations are well established in those areas. In addition, dengue is noticeable by its absence in these areas. 

The issue I have with this work is that it really is based on modeling projections that were conducted five to six years ago. Indeed the models were based on input up to 2015, and so are somewhat dated. I was hoping that the paper would include a benchtop exercise / literature review of the known and current areas with both vector and dengue transmission. The Aedes model makes some embarrassing whiffs, showing Ae. aegypti widespread across northern Australia (it is limited to coastal northern Queensland), and missing the populations in the southwestern United States. The dengue model is likewise based on projections, with dengue probable in much of the SE USA despite a paucity of transmission there. 

What are we to make of a study that is the product of 2 less than perfect models? Is it error squared? It certainly does not reflect the title of “Global patterns…” that indicate real distributional data. So I think that you need to describe this work as a modeling exercise, an explorative dive into potential links between vector, virus and the environment.

I also feel that the paper needs more thorough discussion of reasons why dengue may not occur in areas where Ae. aegypti does. This would be a function of, as you did point out, the development and wealth of the population such that they have access to piped water, window screening and air conditioning. This of course is brought out in the Reiter 2003 publication that you did cite but perhaps a brief discussion of the role that Ae. aegypti as an indoor mosquito has in urban dengue transmission. Other aspects include herd immunity, access to endemic cases, insecticide resistance. Other aspects that you should discuss is that dengue occurs in areas where there are no Ae. aegypti, see outbreaks in southeast China and Japan that have been vectored by Ae. albopictus. 

A few other suggestions:

1. The colours/shades on the map do not work well. I have a very hard time seeing the pale yellow. I suggest seeking the following website: (https://colorbrewer2.org/#type=sequential&scheme=BuGn&n=3). And revising.

2. Please state what sort of populations, mosquito or human, when used.

Reviewer #2: Olson et al. present results on development of a global map detailing probability estimates of Aedes aegypti occurrence and likelihood for dengue transmission. Overall, the manuscript is well-written and presents results that should be of interest to local/regional health authorities as they look to control arbovirus transmission. I only have some minor comments for the authors to consider:

Lines 78-81: these lines highlight a general pattern in the manuscript to alternate between the use of “dengue” and “DEN” – I’d suggest picking one and being consistent.

Lines 138-143: it seems important to justify up front why the choice was made to combine monthly information on temperature and precipitation into annual mean temperatures and cumulative rainfall. This seems to be an overly simplistic approach that ignores the known effects on basic biology and population dynamics, e.g., maximum/minimum temperatures and timing and amounts of rainfall. As the authors indicate (lines 255-264) that choice likely impacted the predictability of their results.

PLOS authors have the option to publish the peer review history of their article (what does this mean?). If published, this will include your full peer review and any attached files.

Reviewer #1: No

Reviewer #2: No
---

## [Decision Letter · Decision Letter 1]

19 Apr 2021

Dear Dr. Hamer,

We are pleased to inform you that your manuscript 'Global patterns of aegyptism without arbovirus' has been provisionally accepted for publication in PLOS Neglected Tropical Diseases.

Best regards,

Mariangela Bonizzoni

Associate Editor

Pedro Vasconcelos

Deputy Editor

Reviewer's Responses to Questions

**Key Review Criteria Required for Acceptance?**

**Methods**

-Are the objectives of the study clearly articulated with a clear testable hypothesis stated?

-Is the study design appropriate to address the stated objectives?

-Is the population clearly described and appropriate for the hypothesis being tested?

-Is the sample size sufficient to ensure adequate power to address the hypothesis being tested?

-Were correct statistical analysis used to support conclusions?

-Are there concerns about ethical or regulatory requirements being met?

Reviewer #1: Approoriate

Reviewer #2: (No Response)

**Results**

-Does the analysis presented match the analysis plan?

-Are the results clearly and completely presented?

-Are the figures (Tables, Images) of sufficient quality for clarity?

Reviewer #1: Thanks for improving the map

Reviewer #2: (No Response)

**Conclusions**

-Are the conclusions supported by the data presented?

-Are the limitations of analysis clearly described?

-Do the authors discuss how these data can be helpful to advance our understanding of the topic under study?

-Is public health relevance addressed?

Reviewer #1: Yes, and thanks for attending to my suggestions.

Reviewer #2: (No Response)

**Editorial and Data Presentation Modifications?**

Reviewer #1: fine

Reviewer #2: (No Response)

**Summary and General Comments**

Reviewer #1: Most of my concerns were addressed. Thank you.

Reviewer #2: The author's have responded appropriately to my previous comments and I support publication of the revised manuscript.

PLOS authors have the option to publish the peer review history of their article (what does this mean?). If published, this will include your full peer review and any attached files.

Reviewer #1: No

Reviewer #2: No

---

## [Editor Report · Acceptance letter]

30 Apr 2021

Dear Dr. Hamer,

We are delighted to inform you that your manuscript, "Global patterns of aegyptism without arbovirus," has been formally accepted for publication in PLOS Neglected Tropical Diseases.

Best regards,

Shaden Kamhawi

co-Editor-in-Chief

Paul Brindley

co-Editor-in-Chief
